# The Association Between Vascular Inflammation and Depressive Disorder. Causality, Biomarkers and Targeted Treatment

**DOI:** 10.3390/ph13050092

**Published:** 2020-05-12

**Authors:** Hans O. Kalkman

**Affiliations:** Retired pharmacologist, Gänsbühlgartenweg 7, CH-4132 Muttenz, Switzerland; hans.kalkman@bluewin.ch; Tel.: +41-61-362-0110

**Keywords:** systemic low-grade inflammation, oxidative stress, geriatric depression, gender differences, leptin, EKODE, fish-oil

## Abstract

Diabetes, obesity, atherosclerosis, and myocardial infarction are frequently co-morbid with major depressive disorder. In the current review, it is argued that vascular inflammation is a factor that is common to all disorders and that an endothelial dysfunction of the blood-brain barrier could be involved in the induction of depression symptoms. Biomarkers for vascular inflammation include a high plasma level of C-reactive protein, soluble cell-adhesion molecules, von Willebrand factor, aldosterone, and proinflammatory cytokines like interleukin-6 or tumor necrosis factor α. A further possible biomarker is flow-mediated dilation of the brachial artery. Treatment of vascular inflammation is expected to prevent or to reduce symptoms of depression. Several tentative treatments for this form of depression can be envisioned: eicosapentaenoic acid (EPA), valproate, Vagus-nerve stimulation, nicotinic α7 agonists, and agonists of the cannabinoid CB_2_-receptor.

## 1. Introduction

Although depression is considered a disorder of the central nervous system, numerous peripheral inflammatory diseases are associated with depressive symptoms [1,2,3,4,5,6]. The presence of elevated levels of proinflammatory cytokines in blood of patients with major depressive disorder has been confirmed in multiple meta-analyses [7,8,9]. Therefore, the idea that inflammatory processes outside of the brain can contribute to the pathogenesis of the major depressive disorders, has gained support during recent years [10,11,12,13]. In particular, cardiovascular disorders like atherosclerosis [14], diabetes [15,16], coronary heart disease [17,18], myocardial infarction [19], or congestive heart failure [20,21] and their risk factors like tobacco smoking [22], obesity [23,24,25], homocysteinemia [26], or old age [27,28] involve vascular inflammation and display comorbidity with depression. For this reason, it is deemed useful to explore the causal link between vascular inflammation and depression, since it may offer information about specific biomarkers and possibilities for targeted treatment. 

## 2. Causes of Vascular Inflammation

### 2.1. Comorbidity Between Diabetes and Depression

Diabetes is characterized by hyperglycemia and insulin resistance in adipose tissue [29]. Glucose metabolism in endothelial cells during hyperglycemia induces a chronic elevation of intracellular diacylglycerol, which provokes a protein kinase C-dependent production of superoxide. This reactive oxygen species promotes vascular inflammation by the activation of NFκB-mediated transcription of the adhesion molecules VCAM1 (vascular cell adhesion molecule-1) and ICAM1 (intercellular cell adhesion molecule-1), and the chemokines MCP1 (monocyte chemotactic protein-1) and IL8 (interleukin-8) [29]. Hyperglycemia also promotes a chemical reaction between the aldehyde-group of sugars and the amino group of amino acids [30]. These products are called ‘advance glycation end products’ (or ‘AGE’). AGEs represent a quite complex class of covalently-modified proteins that are formed by a non-enzymatic reaction between an aldehyde (mostly from sugar) and amine groups from proteins [31]. Such reaction products exert a pathogenic role in disorders that involve an oxidative process, like diabetes or atherosclerosis [31]. AGEs exert their tissue-damaging and proinflammatory effects via activation of a receptor that is referred to as ‘RAGE’ and which is expressed by immune cells, endothelial cells, and vascular smooth muscle cells [30]. Activation of RAGE on endothelial cells leads to enhanced activity of NFκB and, as a consequence, transcription of adhesion molecules [30,32]. Insulin resistance in adipocytes leads to the release of free fatty acids into the circulation [29]. Free fatty acids act as toll-like receptor-4 (TLR4)-agonists and trigger NFκB-mediated production of the proinflammatory cytokines, TNFα and IL6 [33]. Moreover, in hepatocytes these cytokines stimulate the release of acute-phase protein, CRP (C-reactive protein), whereas TNFα, IL6 and CRP can, in turn, provoke insulin-resistance in target cells such as the endothelium [29,34]. Hepatocytes filter the circulating free fatty acids and pack them into very-low density lipoprotein (VLDL) particles. VLDL particles, especially when containing oxidized VLDL are known to contribute to endothelial dysfunction [29]. So, diabetes activates circular processes that result in enhanced leukocyte diapedeses, platelet reactivity, mitogenesis, blood coagulation, and vasoconstriction, while endothelium-dependent vasodilatation is impeded [34]. Notably, these deleterious vascular effects also occur at the blood brain barrier (BBB). For instance, there is a significant association between diabetes on the one hand and reduced cerebral blood flow or lacunar infarcts on the other hand [35]. So, diabetes increases several factors that are associated with vascular inflammation. Large meta-analyses confirm the high comorbidity of diabetes and depression [15,16]. 

### 2.2. Comorbidity Between Female Obesity and Depression

Overnutrition is an essential cause of insulin resistance and leads to obesity and atherosclerosis [36]. In prospective cohort studies, the association between obesity and depression was determined to be bidirectional [24,25]. Obesity rates are high in depressed patients, and conversely, the incidence of depression in obese individuals approaches a remarkable 30% [23]. Notably, from cross-sectional studies, it seems that obesity is associated with depression in females, but this is apparently not the case for male obesity [25,37,38,39]. This gender difference may relate to a difference in leptin signaling. Leptin is produced by adipocytes from white fat, and over-nutrition raises the circulating levels of leptin [40]. Indeed, obese depression patients display high levels of leptin in plasma [23,24,39]. Reduction in body fat by means of dietary restriction provoked not only a reduction in leptin levels but in parallel also led to a reduction in depression scores [41]. But importantly, whereas high leptin levels seem to predict depression ratings in females, they did not associate with male depression [39]. It should be noted that comorbidities like hypertension, hyperlipidemia, or inflammation parameters like plasma CRP, IL6, or TNFα did not differ between genders [39]. Activation of leptin receptors on human abdominal perivascular adipocytes [42] or human adrenal cortex [43,44] stimulates aldosterone production. In females, visceral obesity directly correlates with plasma aldosterone levels, independently of plasma renin-levels [43,45]. Weight loss with a concomitant reduction of visceral adipose tissue in females led to a reduction in plasma aldosterone levels. In males, however, such an effect was not observed [45]. These data indicate that in females, but not in males, leptin raises plasma levels of aldosterone, and this latter factor may be causal for vascular inflammation and depression symptoms (see below). 

### 2.3. Aldosterone as a Risk Factor for Depression

Data from clinical studies suggest that an elevated level of aldosterone is associated with an increased risk for depression. Male and female depression patients exhibited significantly higher aldosterone levels than matched controls [46,47]. Moreover, in patients with premenstrual syndrome, increases in plasma aldosterone during the late luteal phase were significantly associated with increases in emotional and physical symptoms [48,49]. Spironolactone, a mineralocorticoid-receptor antagonist, relieved the somatic and psychological symptoms of the premenstrual syndrome [48,50]. Together with data from the previous paragraph, it seems that obesity in females increases, through activation of the leptin receptor, the circulating levels of aldosterone. It is presumably aldosterone that ultimately enhances the depression risk. Concerning the mechanism of action, it is known that acute infusion of aldosterone into the human forearm inhibits endothelium-dependent vasorelaxation to acetylcholine [51]. Endothelium-dependent dilatory response to acetylcholine was also impaired in mice with genetic or diet-induced obesity [42,52]. In these latter studies, the vasodilator response was ‘rescued’ by pretreatment with the mineralocorticoid receptor (MR) antagonist eplerenone, or by endothelium-specific deletion of the MR-gene [52]. In rat cerebral arteries, aldosterone increased superoxide production and raised the transcription of chemokines from endothelial cells [53]. Incubation of endothelial cells from the human umbilical vein with aldosterone increased mRNA and protein levels of the adhesion molecules ICAM1, VCAM1, and selectin-E, but reduced the production of the endothelial vasodilator, nitric oxide [54,55,56,57]. These results indicate that aldosterone, via activation of endothelial mineralocorticoid receptors, promotes a proinflammatory state of the endothelium [43,44,58,59]. Faulkner et al. [59] have reported that progesterone increases the expression of aldosterone-receptors on endothelial cells. This observation provides a possible explanation of why females are more sensitive to obesity-, leptin- and aldosterone-induced depression than males. 

Apart from leptin, there are some additional aldosterone-releasing factors [60]. Typical for West European and North American diets is the consumption of large amounts of the vegetable fatty acid, linoleic acid. This 18-carbon, ω-6 unsaturated fatty acid is oxidized to the compound ‘12,13-epoxy-9-keto-10(trans)-octadecenoic acid’ or ‘EKODE’ [61]. EKODE (see Figure 1) is a particularly strong activator of aldosterone release from adrenal cells and is detected in substantial amounts in human plasma [60,61]. In blood samples from 24 adults, levels of EKODE correlated with those of aldosterone [61]. In a prospective study in American nurses, a reduction in linoleic acid-intake was associated with a reduction in depression risk [62,63]. High EKODE levels are also present in itching psoriatic skin lesions [64]. Consistent with an increase in plasma aldosterone levels [65], psoriasis is associated with ischemic heart disease, cerebrovascular disease, peripheral artery disease [66], hypertension [67], and depression [68]. A link between EKODE and depression has also been provided by a study of Hennebelle et al. [69] in patients with seasonal affective disorder. In this study, EKODE levels were increased during the dysthymic period but were back to normal during the euthymic phase [69]. Whether also the aldosterone levels were elevated during the seasonal affective period was not investigated, unfortunately.

### 2.4. Old Age as a Risk Factor for Depression

Comorbidity between cardiovascular disease and depression in the elderly has been recognized for several decades and has led to the formulation of the ‘vascular depression’ hypothesis [70,71]. This hypothesis proposes that ‘cardiovascular disease may predispose, precipitate, or perpetuate some geriatric depressive syndromes’ [70]. Geriatric depression is sometimes referred to as late-life depression [27] or as late-onset depression [28], depending on the inclusion/omission of elderly depressed patients who suffered from depression before the age of 40. In the longitudinal study by Van Agtmaal et al. [27], the presence of white matter hyperintensities and cerebral microinfarctions was associated with depression symptoms in patients older than 40 years of age. This result was corroborated by the meta-analysis by Salo et al. [28], which confirmed that the white matter hyperintensity-burden was particularly pronounced in patients with late-life depression. White matter hyperintensities, seen in magnetic resonance imaging brain scans, are assumed to reflect silent lesions of vascular or ischemic origin in the brain [72]. Such hyperintensities, in principle, can also occur in young patients, for instance, in those at elevated risk for atherosclerotic disease [73,74]. 

### 2.5. Homocysteine and Vascular Inflammation

High plasma levels of homocysteine are associated with atherosclerosis (for review see [75]) and with male major depressive disorders [26]. The molecular basis for the activity of homocysteine remains undefined, however, several mechanisms that perturb endothelial function have been identified. These include reduced nitric oxide production, enhanced superoxide formation, loss of endothelial glutathione, activation of endothelial NFκB, transcription of MCP1, IL8, ICAM1, and VCAM1, and an increase in pro-thrombotic platelet function [75]. The gender-specific effect of homocysteine levels on depression may be explained by a protective effect of estrogen on endothelial integrity [76].

### 2.6. The Endothelium of the Blood-Brain Barrier

The endothelial layer of cerebral arterioles constitutes one of the physical barriers between the blood circulation, including any circulating proinflammatory cytokines and the brain [77,78]. It is conceivable that endothelial dysfunction, as caused by vascular inflammation, is the underlying factor that explains why depression and cardiovascular disorders co-occur [79,80,81,82]. Dysfunction of the endothelium is also implicated in disease states like hypercholesterolemia and other dyslipidemias, diabetes, obesity, hypertension, and aging [83]. Endothelial cell dysfunction, including that of the blood-brain barrier, is associated with an increased adherence of leukocytes, platelet activation, stimulation of the coagulation cascade, induction of an inflammatory environment, and formation of atherosclerotic plaques [57,84]. It is conceivable that such processes lead to an altered function of the underlying brain tissue, although a detailed understanding of the pathological intricacies is lacking.

## 3. Biomarkers for Vascular Inflammation

### 3.1. Flow-Mediated Dilation

Endothelial dysfunction can be estimated non-invasively by a technique referred to as ‘flow-mediated dilation’ (FMD) [85,86]. Deficits in FMD in peripheral blood vessels are associated with depression symptoms [27,87]. In fact, blunted FMD responses correlate both with the presence and the severity of depression symptoms [79,88,89,90]. These results make FMD a useful biomarker for depression that is secondary to a vascular inflammatory disorder.

### 3.2. Circulating Biomarkers

Endothelial inflammation involves activation of the nuclear factor-κB (NFκB) pathway, which in turn, is triggered by multiple factors including C-reactive protein (CRP), cytokines like tumor necrosis factor-α (TNFα) and interleukin-6 (IL6), RAGE-agonists and molecules derived from pathogenic microorganisms. The NFκB pathway promotes transcription of molecules that increase leukocyte-adhesion, such as ICAM1 and VCAM1 [29,91,92,93,94]. Circulating levels of CRP, TNFα, and IL6 are increased both in patients with atherosclerosis and in patients with depression (see Table 1). Proteolytic cleavage of membrane-bound ICAM1 and VCAM1 results in soluble forms (sICAM1 and sVCAM1, respectively). In soluble form, these products continue to act as proinflammatory agents [95,96], and their increased levels are predictive for the later atherosclerotic disease [96,97,98,99]. Therefore, plasma levels of sICAM1 and sVCAM1 might be useful biomarkers for depression associated with vascular inflammation (see [100,101]). The von Willebrand factor (vWF) is a further factor that is implicated in the pathogenesis of atherosclerosis. The vWF is involved in macrophage and leukocyte recruitment to inflamed blood vessels [102], in platelet adhesion in stenotic arteries [103], and importantly, is linked to stroke and impairment of the blood-brain barrier function [104,105,106]. High circulating levels of vWF are seen in myocardial infarction [107], acute coronary artery disease [108], and carotid stenosis [103]. Consistent with the hypothesis that depression can be secondary to vascular inflammation, plasma vWF levels of patients with major depression, independent on their antidepressant treatment, were significantly higher than those of healthy control subjects [82,109,110,111].

## 4. Treatments

Data reviewed above indicates that vascular inflammation deteriorates endothelial function, including the blood-brain barrier, and this, presumably, leads to depression. Consequently, treatments that prevent or improve endothelial dysfunction could result in antidepressant effects in patients with this particular cause of depression. 

### 4.1. The Fish-Oil Component Eicosapentaenoic Acid (EPA)

Zehr and Walker [132] reviewed 22 clinical trials, in which the effect of fish-oil supplementation on endothelium-dependent vasodilation was investigated. In 18 out of these 22 studies, an improved vasodilator effect was found. This included studies in hyperlipidemic individuals, in young cigarette smokers, in patients with type-2 diabetes, and individuals with a high body mass index [132]. The two major constituents of fish-oil are the 20-carbon ω3-fatty acid eicosapentaenoic acid (EPA) and the 22-carbon ω-3 fatty acid docosahexaenoic acid (DHA) (for review see [133,134]). Supplementation of the diet with fish-oil rapidly increases free circulating levels of EPA and DHA, as well as their levels in membranes of erythrocytes, platelets, and monocytes [133,135]. But, whereas DHA is particularly abundant in human retina and brain [135,136,137], the concentration of EPA in these CNS-tissues is some 250-times lower than that of DHA, despite the fact that EPA and DHA enter the brain at similar rates [137,138]. DHA is rapidly incorporated into brain phospholipids to maintain high levels [136], whereas EPA is slowly incorporated and extensively metabolized [138]. These data suggest that DHA is important for brain function, while EPA most likely will have its main function outside the CNS. 

Apart from fish-oil, impaired endothelium-dependent blood vessel relaxation was also improved by supplementation with pure EPA [132]. In a prospective cohort study in adults with congestive heart failure, EPA-concentrations, but not DHA-concentrations, were inversely associated with incident coronary heart failure [139]. In a multi-ethnic sample of nearly 3000 adults with atherosclerosis, high EPA plasma levels (or combined DHA and EPA levels) were associated with lower sICAM1 levels [140]. Moreover, EPA supplementation, either as fish-oil or as pure product, has been reported to decrease circulating levels of the adhesion molecules sICAM1 and sVCAM1 [62,141,142]. This effect was noted in both subjects with dyslipidemia and in healthy controls [141]. In contrast, pure DHA failed to affect sICAM1 levels [142]. Large meta-analyses consistently report that the antidepressant activity of EPA is superior to that of DHA [143,144,145,146], whereas pure DHA failed to reduce depression [147]. Yang et al. [148] reported a head-to-head comparison of the antidepressant activity of pure EPA and DHA. In this 12-week double-blind, randomized controlled trial, participants with major depressive disorder were randomly assigned to receive EPA, DHA, or a combination of EPA and DHA. The cumulative rates of clinical remission were significantly higher in the EPA and the EPA + DHA group than in the DHA group. Clinical remission correlated with the plasma levels of the EPA-derived endocannabinoid metabolite EPEA (eicosapentaenoyl ethanol-amide; see Figure 1), but not with those of any other endocannabinoid [148]. This observation indicates that the antidepressant effect of EPA might be due to a transformation of EPA to an endogenous cannabinoid metabolite.

### 4.2. CB_2_-Receptor Agonists

Endocannabinoids are an important class of biologically active fatty acid-derivatives [136,149]. They play a role in the cardiovascular system [150], including blood vessels of the blood-brain barrier [151,152], in immune function [137,153,154], and in depression [10,155]. Perhaps the best-studied endocannabinoid is anandamide (arachidonyl ethanolamide, or AEA), an amide formed from ethanolamine and arachidonic acid. Similar to arachidonic acid, cells such as adipocytes are able to convert DHA to DHEA (docosahexaenoyl ethanolamide) and EPA to EPEA [156]. These compounds activate the classical endocannabinoid CB1 and CB2 receptors, and several orphan receptors, including GPR18 [156,157]. Since phytocannabinoids strongly affect mental function, it is conceivable that a receptor-subtype selective compound might display a beneficial effect in major depression. However, no such compound has been introduced to the market.

Similar to free polyunsaturated fatty acids, cyclooxygenase (COX), lipoxygenase (LOX), and cytochrome P450 (CYP) enzymes also oxidize endocannabinoids [137,158,159,160]. CYP ω-3 epoxidases oxidize EPEA and DHEA to EEQ-EA (Figure 1) and EDP-EA, respectively [160]. These anti-inflammatory products were observed in significant amounts in the brain and peripheral tissues of the rat [160]. Moreover, EEQ-EA and EDP-EA exert vasodilator activity in bovine coronary arteries and inhibit platelet aggregation [160,161]. Epoxidation of AEA and EPEA greatly altered the CB1/CB2 profile by causing a profound shift towards higher CB2 affinity [160]. In contrast, the affinity of DHEA and its ω-3 epoxidation product EDP-EA for CB1 and CB2 receptors was 100 to 1000-fold less than that of the arachidonic acid and EPA derivatives [160]. Thus, the DHA-derived oxidized endocannabinoid EDP-EA failed to bind to CB2-receptors, whereas the equivalent EPA-derived mediator (EEQ-EA) was highly potent and selective for CB2-receptors. This represents a major difference between the EPA- and DHA-derived mediators and potentially could explain the difference in antidepressant activity since there is ample evidence that stimulation of endothelial CB2 receptors counteracts increases in ICAM1 and VCAM1 [162,163,164]. The beneficial role of CB2-receptor agonists in endothelial dysfunction [165] could result in antidepressant activity in depression caused by vascular inflammation. 

### 4.3. Vagus Nerve Stimulation

Stimulation of the left cervical Vagus nerve is an approved therapy for ‘treatment-resistant’ depression [166,167,168]. The cholinergic anti-inflammatory pathway [169] is part of a neural reflex circuit that detects peripheral inflammation and provides regulatory feedback to suppress an excessive activation of the innate immune system [170,171]. The efferent limb of the Vagus nerve provides parasympathetic innervation to multiple organ systems, including the vascular endothelium [172,173]. An important cholinergic receptor-subtype that is activated by Vagus nerve- stimulation is the nicotinic receptor α7 (α7 nAChR) [174]. In endothelial cells from human umbilical vein, stimulation of α7 nACh-receptors by nicotine or acetylcholine inhibited the activation of the NFκB pathway and transcription of proinflammatory cytokines and chemokines [172]. Moreover, stimulation of endothelial α7 nACh-receptors reduced inflammation-induced expression of VCAM1, ICAM1 and E-selectin, and the subsequent trans-endothelial migration of leukocytes [172]. In rodent models, activation of α7 nACh-receptors improved atherosclerosis [175], triglyceride levels, weight gain, metabolic syndrome [176,177], and heart rate variability [178]. Collectively, these data indicate that Vagus nerve-stimulation and α7 nACh-selective agonists could be useful in the treatment of depression that is secondary to vascular inflammation. 

### 4.4. Histone Deacetylase inhibition

The clearance of apoptotic neutrophils by monocytes/macrophages is a crucial aspect of atherosclerosis since its failure hampers the resolution of vascular inflammation [179]. Whereas inflammatory molecules like LPS (lipo-polysaccharide) or TNFα delay the apoptosis of aged neutrophils [180], histone deacetylase inhibitors (e.g., short-chain fatty acids like propionate, butyrate, or valproate) increase the apoptosis of neutrophils [180]. Valproate and butyrate increased the clearance of apoptotic neutrophils by macrophages and reduced zymosan-induced peritonitis in mice [181]. These histone deacetylase (HDAC) inhibitors were found to suppress the LPS-induced release of IL6 and TNFα [181]. Interestingly, propionate and butyrate are generated through microbial fermentation of otherwise indigestible carbohydrate fibers [12], providing an explanation for the notion that fiber-rich food is good for health [182]. Valproate significantly reduced atherosclerosis in laboratory animals [183,184], whereas valproate and butyrate displayed anti-inflammatory and neuroprotective effects in an animal model of stroke [185]. In addition, genetic disruption of the HDAC3 gene in endothelial cells from human umbilical vein, inhibited NFκB-signaling, reduced the expression of VCAM1 and suppressed monocyte adhesion [186]. Consistent with the preclinical data, in epidemiological studies, valproate seems to be effective for primary and secondary prevention of stroke [187,188] and for myocardial infarction [189]. Although not thoroughly investigated, available data indicate that valproate possesses antidepressant activity too [190,191,192]. Based on this information, HDAC inhibition could represent a specific treatment for vascular inflammation-induced depression. 

## 5. Discussion

Although many depressed patients display elevated levels of proinflammatory cytokines, this is not the case in every patient [116,193,194]. If we assume that all forms of depression are due to inflammation, then in some patients, the inflammation is not traceable in the blood. The alternative is that there are forms of depression that are unrelated to inflammatory events (but in that case, there is currently no clue about the cause of this form of depression). The present review has focused on depression associated with elevated levels of proinflammatory cytokines in the circulation, a form of depression that has been described as ‘inflammatory cytokine associated depression (ICAD)’ [4]. In this subgroup of depression patients, the peripheral inflammation took place before the onset of depression symptoms [5,8,116]. The focus on this subtype is considered justifiable, since high levels of IL6, soluble IL-6 receptor (sIL6R), or TNFα generally indicate a poor response to conventional antidepressants [8,117,195,196,197]. Notably, a poor response to antidepressant treatment occurs in 30–50% of patients with major depressive disorder [198]. In the case of chronic non-resolving disorders like autoimmune diseases, but also for lifestyle and diet-related disorders like obesity, type-2 diabetes, or smoking-induced atherosclerosis, it is likely that the poor response to classical antidepressants is due to the persistence of the inflammatory process. Lotrich [4] suggested that depression secondary to vascular inflammation might be more responsive to other medications than to classical antidepressants. As the mechanism of action of the treatments proposed in the present review is complementary to the central effects by traditional antidepressants, it is expected that they will augment the clinical response to current treatment (see Figure 2). 

Presently, terms like ‘ICAD’ or ‘vascular depression’ are not generally accepted and are not entities acknowledged in disease classification systems such as DSM or ICD. Similarly, the therapeutic use of valproate, EPA, Vagus nerve-stimulation in depression is currently not firmly established. It is the author’s opinion that progress in the treatment of major depression is achievable if we would recognize that depressive symptoms occur as a consequence of vascular inflammation. For this purpose, the current review has proposed a number of biomarkers. These biomarkers would enable stratification of depressed patients into those with and those without vascular inflammation. Rapaport et al. [199] have published a motivating example of this approach. They tested pure EPA and pure DHA and obtained evidence for a therapeutic effect of EPA in depressed patients with high levels of peripheral inflammation markers, while without patient stratification, the effect of EPA would have been missed. 

## Figures and Tables

**Figure 1 pharmaceuticals-13-00092-f001:**
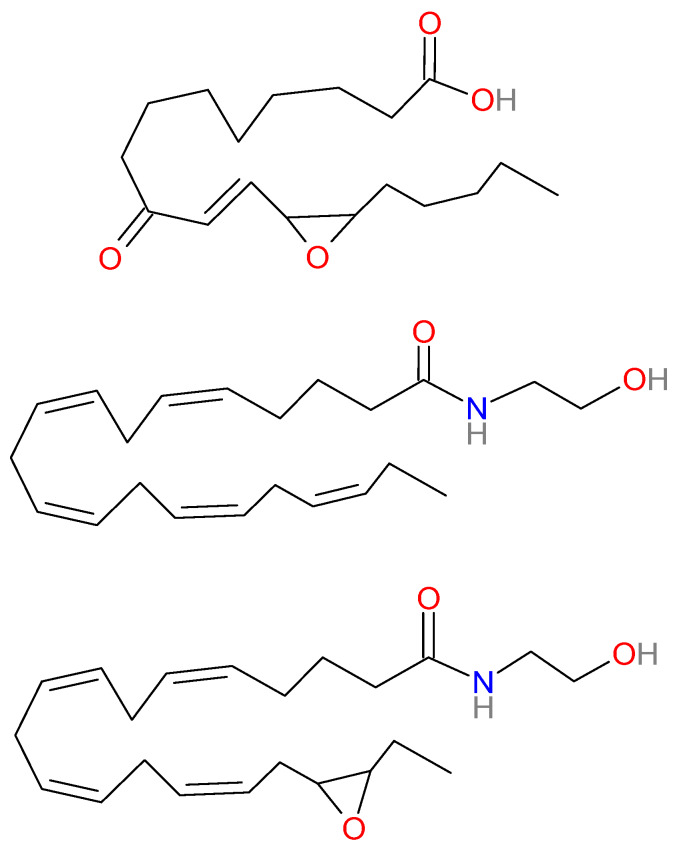
From top to bottom are shown the chemical structures of EKODE, EPEA, and 17,18-EEQ-EA.

**Figure 2 pharmaceuticals-13-00092-f002:**
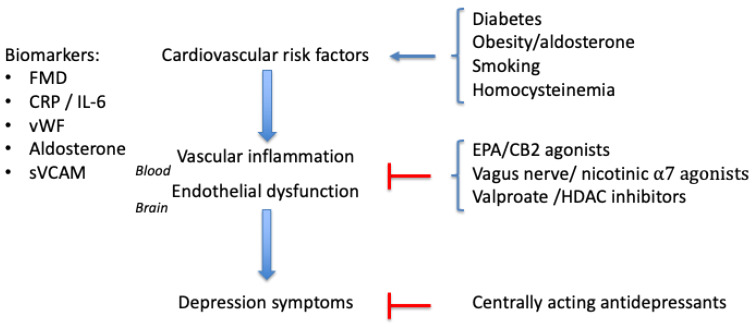
Diabetes and other risk factors for atherosclerosis promote vascular inflammation and endothelial dysfunction. Biomarkers for this process are FMD (flow-mediated dilation), CRP (C-reactive protein), proinflammatory cytokines like interleukin-6, von Willebrand factor (vWF), aldosterone, or soluble vascular cell adhesion molecule (sVCAM). Putative treatments specific for depression secondary to vascular inflammation are eicosapentaenoic acid (EPA), cannabis CB2 receptor agonists, Vagus nerve stimulation, agonists for the nicotinic α7 receptors, and HDAC (histone deacetylase) inhibitors, like valproate.

**Table 1 pharmaceuticals-13-00092-t001:** Potential Biomarkers for Depression Caused by Vascular Inflammation.

Factor	Vascular Inflammation	Major Depressive Disorder
CRP	[112,113,114,115]	[116,117,118,119,120]
IL6	[115,121,122]	[123,124]
TNFα	[125]	[126,127]
sICAM1	[96,99]	[82,100,101,128,129,130]
sVCAM1	[97,98]	[82,100,101,129,130]
vWF	[103,104,105,106]	[82,109,110,111]
Aldosterone	[43,58,59,131]	[46,47,49]
Homocysteine	[75]	[26]

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
