# Peer review of "The Association Between Vascular Inflammation and Depressive Disorder. Causality, Biomarkers and Targeted Treatment"

_pharmaceuticals, 2020, doi:10.3390/ph13050092_

Round 1

Reviewer 1 Report

Manuscript ID: pharmaceuticals-796271

Title: The association between vascular inflammation and depressive disorder. Causality, biomarkers and targeted treatment.

Reviewer’s Comments:

Thank you for the opportunity to review this manuscript. The author made an appreciable attempt to review the relation between vascular inflammation and depression using existing well-known markers, suggesting the treatments. The study is pleasant to read and presents the results concisely. Let me share some minor concerns and comments with you:

Keywords

Some more words may be necessary. Ex. Chronic inflammation or systemic low-grade inflammation, oxidation, etc.

Chapter 2  In the second chapter, author mentions the associations between vascular inflammation and depression.

L.44-45

I suggest to explain the AGE in a little more detail for readers. In addition, I found an interesting article presenting mental condition and an AGE:

  J Psychosom Res. 2019; 126: 109825.

  Arch Physiol Biochem. 2019; 15: 1-9

L.51-53

If you mention “oxidized” VLDL, it would become this review more persuasive.

L.56

Blood-brain barrier should be abbreviated as BBB. It’s general.

L.67-

It is very important to analyse by gender in psychosomatic medicine. I agree your point of view.

L.160-

Author describes the endothelium function of the blood-brain barrier (BBB). I suggest to explain in a little more detail concerning the vulnerable BBB. The following article presents the inflammatory signaling into the brain.

    Pharmacol Ther. 2011; 130: 226-238

Chapter 3

L.197-

IL-1 (α and β), IL-8, IL-17, IL-23 etc. also increasing vascular inflammation. It may be better to add some more inflammatory factors in to Table 1.

Chapter 4

I appreciate the discussion concerning the differences the EPA and DHA function.

Author explains the cannabinoid (CB). I recommend to explain how the CBs affect our psychological status.

Furthermore, there might be a risk of possible other variable explanations which could be discussed (e.g. SLE, atopic dermatitis, asthma, etc.). However, this manuscript provide us significant information to treat or prevent depression.

Author Response

I am grateful to referee 1 for his pleasantly voiced, constructive criticisms and improvement proposals. I have followed nearly all his proposals (see manuscript: in red color; note that the numbering of the citations has changed because of the amendments). From the two papers on AGE that he mentioned, I could identify only the first one (the second must contain a typing error; I have gone through the articles on the web-site of the journal, but I still don’t know, which one the referee means). The first paper is on the influence of green tea on the general health. This is somewhat too far outside of the topic of my review.

The second point on where I do not follow the referee’s proposal, is on the inclusion of further biomarker-candidates in table 1. I am of course fully aware of data on IL1alpha, IL8 etc. I have cited several meta-analyses that discuss these markers, but these meta-analyses often exclude these additional markers, either because of a small data set, or lack of replication. It was not my goal to be totally comprehensive. Rather the point that I want(ed) to emphasise in the manuscript is, that vascular inflammation may constitute a peripheral cause for depression.

It is true that there are many other peripheral causes of inflammation that also are co-morbid with depression. In an earlier version of the manuscript, I had a large table including SLE (and Th2-inflammatory disorders like atopic dermatitis and asthma!). But then, I felt that this diluted the message and unduly extended the size of the manuscript. So, although I fully agree with the referee, I have not re-introduced this table. 

Finally, it is great to see that the referee recognises the importance of gender differences. Actually, I think that major journals should urge authors to report their clinical data in a form that allows the testing of gender differences. I predict that many unexpected findings will emerge.       

Reviewer 2 Report

In the present paper the Author aimed to explore the causal link between vascular inflammation and depression, as it may offer information about specific biomarkers and possibilities for a targeted treatment.

Overall, I found this paper timely, well written, very interesting and scientifically sound. I have some suggestions aimed to improve the high quality of the paper and these are outlined below:

1) I suggest the Author to add how the literature for this review was searched and if there were inclusion or exlusion criteria.

2) I believe that the table 1 is useful. However, I would suggest, if feasible, a figure depicting the link between these biomarkers and how they interact in both vascular inflammation and depression.

3) Concerning CRP levels there are several studies that should be briefly mentioned with appropriate references (see De Berardis et al. CNS Spectr. 2017, Int J Immunopathol Pharmacol. 2008 and Int J Immunopathol Pharmacol. 2006) .

4) Concerning treatments, I believe that the role of newer antidepressant should be briefly discussed.

Author Response

I would also like to thank referee 2 for the time he spent reviewing my manuscript, as well as for the constructive comments that he made. Especially the paper about the effect of agomelatine on CRP has slipped my attention.

My literature search is very archaic. I have read the literature on depression since decades (during the time that I worked in the preclinical department of Novartis, and thereafter during my retirement), and for each paper that I read I have made a small summary (in Word). In this file, over the years, a substantial amount of information has accumulated, and it allowed me to simply check on information once I would have a novel thought. So, I did not use a formal literature search and strategy. The consequence is that I undoubtedly will have missed papers, but on the other hand, there will be relevant references and insights in my manuscript that would have missed by a formal search. The notion that atherosclerotic diseases may represent a peripheral cause for depression, is currently somewhat neglected, and (by some) perhaps not even recognised. This is the message that I have tried to convey, and I have used the technology that is still available to a retired pharmacologist.

On point 2, I also have to disappoint the referee. Actually, I have indeed been thinking about a cartoon that would logically connect the different biomarkers. But it would be an oversimplification, and in the end, it would overstate the strength of currently available information.

As mentioned, I have missed the papers by De Berardis et al. I have included two of the three citations. The one on bipolar disorder is not quoted, because I wanted to restrict myself to unipolar depression. Agomelatine is a fascinating compound because of its novel mode of action. It has taken me many years to come up with a possible explanation why a mixed 5HT2C antagonist and MT-agonist would act as an antidepressant (Kalkman HO, Feuerbach D. Antidepressant therapies inhibit inflammation and microglial M1-polarization. Pharmacol Ther 2016, 163, 82-93). A tentative explanation is, that the combination of effects results in higher release of IGF-1 during the prolonged slow-wave sleep period. I do not know if this hypothesis is now confirmed or rather refuted. It is conceivable that IGF-1 might have an anti-inflammatory effect on blood vessels. But unfortunately, here I am stapling one hypothesis on top of another. I have to leave this idea to those who can put it to the test. The referee will appreciate, that I am therefor hesitant to write a large discussion on agomelatine (point 4).